# Serological and Molecular Survey of *Leishmania infantum* in a Population of Iberian Lynxes (*Lynx pardinus*)

**DOI:** 10.3390/microorganisms10122447

**Published:** 2022-12-11

**Authors:** Clara M. Lima, Nuno Santarém, Nuno Costa Neves, Pedro Sarmento, Carlos Carrapato, Rita de Sousa, Luís Cardoso, Anabela Cordeiro-da-Silva

**Affiliations:** 1Microbiology Laboratory, Department of Biological Sciences, Faculty of Pharmacy, University of Porto, 4050-313 Porto, Portugal; 2Parasite Disease Group, Institute for Research and Innovation in Health (i3S), University of Porto, 4099-002 Porto, Portugal; 3Institute for the Conservation of Nature and Forests (ICNF), 7800-298 Beja, Portugal; 4Centre for Vectors and Infectious Disease Research (CEVDI), National Institute of Health Dr. Ricardo Jorge, 2965-575 Águas de Moura, Portugal; 5Department of Veterinary Sciences, and CECAV—Animal and Veterinary Research Centre, University of Trás-os-Montes and Alto Douro (UTAD), 5000-801 Vila Real, Portugal; 6Associate Laboratory for Animal and Veterinary Science (AL4AnimalS), University of Trás-os-Montes and Alto Douro (UTAD), 5000-801 Vila Real, Portugal

**Keywords:** ELISA, IFAT, leishmaniasis, *Leishmania infantum*, *Lynx pardinus*, PCR, Portugal, rK39

## Abstract

*Leishmania infantum*, the sand fly-transmitted protozoan parasite responsible for leishmaniasis in humans, dogs, and cats, is endemic in the Iberian Peninsula. However, the impact of *L. infantum* infection on the conservation of the endangered Iberian lynx (*Lynx pardinus*) is unknown. Herein, we describe for the first time the occurrence of *L. infantum* infection among a population of reintroduced and wild-born *L. pardinus* living in the Portuguese Guadiana Valley Park. The presence of infection was addressed by molecular detection of *Leishmania* kinetoplast DNA (kDNA) in 35 lynxes, with further confirmation of *L. infantum* species performed by an internally transcribed spacer (ITS)-1 sequencing. Eight blood samples were positive for kDNA, and ITS-1 sequencing confirmed the presence of *L. infantum* in two of those samples. Exposure to *Leishmania* was screened in a group of 36 lynxes using an immunofluorescence antibody test (IFAT) and a multi-antigen enzyme-linked immunosorbent assay (ELISA), using SPLA, rK39, and CPX as *Leishmania*-specific antigens. Four animals presented a positive IFAT at a dilution of 1:40. Eight samples were considered seropositive to all ELISA *Leishmania*-specific antigens. Agreement between PCR, IFAT, and all ELISA antigens was found for 1 in 27 samples. These results highlight the susceptibility of autochthonous *L. pardinus* to *L. infantum* infection. Further investigation is required to assess the impact of *L. infantum* infection on this wild species conservation.

## 1. Introduction

Leishmaniasis (or leishmaniosis) is one of the most severe neglected tropical diseases affecting the poorest communities of sub-industrialized countries in tropical and sub-tropical regions of the world. It is caused by a dimorphic protozoan parasite of the genus *Leishmania* (Kinetoplastida, Trypanosomatidae) and is widely spread through more than 100 countries within the European, African, Asia, and American continents. The pattern of distribution of parasite species and induced disease manifestations follows the distribution of each *Leishmania* spp. permissive hematophagous phlebotomine vector sand flies. Depending on the infecting species, *Leishmania* have an anthroponotic and/or zoonotic life cycle [1]. 

In the Mediterranean basin, leishmaniasis is caused by *L. infantum*, for which dogs are the primary domestic reservoir. In humans, the parasite is responsible for zoonotic visceral leishmaniasis. Dogs are usually sub-clinically infected [2] but may develop canine leishmaniasis (CanL). Despite the available prophylactic measures, CanL remains a veterinary concern and a public health hazard within endemic regions. The control of CanL relies on preventing exposure to infected female phlebotomine sandflies and on early diagnosis and treatment. The high seroprevalence of CanL within the Mediterranean region [3,4], and the Iberian Peninsula in particular, can be explained by ecological factors, such as the abundance of vectors (mostly *Phlebotomus ariasi* and *P. perniciosus*) [5] and dogs, and by limitations associated with disease and infection management in afflicted dogs. Among these limitations, diagnostic challenges presented by sub-clinically infected animals [6,7] and atypical CanL presentations [8] aggravate the capacity to detect infected animals. This is worsened by laboratory diagnostic limitations [9] and the limited capacity of available therapeutic options to eradicate infection. This scenario is worsened by poor compliance by dog owners, which further limits CanL prevention, diagnosis, and treatment [10]. Over the last 20 years, the emergence of vaccines against CanL have improved the well-being of infected animals, enabling a decrease in disease severity, but this did not disrupt the circulation of parasites in the environment, since vaccines do not offer sterile protection [11]. Importantly, *L. infantum* also thrives outside domestic settings, likely having several sylvatic mammalians as parasite reservoirs [12,13]. The role of cats as reservoirs of *L. infantum* has been increasingly investigated. Since this host species has been found infected and infectious to permissive sandflies [14,15], its abundance and proximity to humans suggests a role in *L. infantum* transmission within urban environments. 

The Iberian Peninsula is the westernmost point of the Mediterranean basin. It is home to one of the world’s most endangered feline species, the Iberian lynx (*Lynx pardinus*) [16]. Currently, the species inhabits restricted areas in Spain (namely Andalusia, Castilla-La Mancha, Extremadura, and Murcia) and Portugal (Guadiana River Valley) [17]. In 2002, to revert the species near extinction, Spanish and Portuguese conservation agencies carried out an ambitious reintroduction project based on captative-bred animals, which allowed the species to recover substantially. In 2014, the first captive-bred Iberian lynxes started being reintroduced into suitable areas within the species’ historical range. Meanwhile, lynx populations have been continuously managed and monitored. By 2021, the number of Iberian lynxes living in the wild reached 1361 animals, including cubs. In the Portuguese reintroduction sites of the Guadiana Valley River Park, 209 individuals, including 70 cubs, have been identified [17]. Despite the success of the reintroduction and the increasing number of wild newborns, factors including climate change, land use, and a reduction in rabbit populations (conditioned by infectious diseases) are still a matter of concern [18]. Throughout the 20th century, non-natural and anthroponotic-related causes of death were largely responsible for the decline of Iberian lynx populations. The most important causes of decline included human persecution, habitat destruction, and prey scarcity [19,20]. On the other hand, neoplasias [19] and infectious diseases [19,21] have been identified as natural causes of morbidity and mortality among this feline species. Several infectious agents have been identified as infecting wild-borne and reintroduced Iberian lynxes. For example, regarding viral infections, feline calicivirus (FCV), feline coronavirus (FCoV), leukemia virus (FeLV), parvovirus (FPV), and feline immunodeficiency virus (FIV) have been diagnosed, as well as canine morbillivirus (CDV), canine adenovirus-1 (CAV-1), and suid herpesvirus-1 [19,21,22]. Parasitic infections such as *Toxoplasma gondii* [19,22] *Cytauxzoon* spp., and *Neospora caninum* [22] have been confirmed, and bacterial agents such as *Chlamydia* spp. [19], *Mycobacterium bovis* [19,21,23,24], *Leptospira interrogans,* and *Ehrlichia* spp. [21] have also been demonstrated. It is now well-established that cats experimentally or naturally infected with *L. infantum* can develop clinical leishmaniasis and produce specific anti-*Leishmania* antibodies [25]. Considering that *L. infantum* is endemic in the natural habitat of the Iberian lynx and that the impact of *Leishmania* infection in these felines is still mostly unknown, it is important to address if Iberian lynxes are as susceptible to infection by *Leishmania*, as domestic cats and other Felidae are [12,13]. Regarding the epidemiology of vector-borne diseases, with particular emphasis on *L. infantum*, these are mainly driven by the presence of competent vectors and also of reservoir hosts, which are responsible for maintaining pathogens within a suitable ecosystem. Nevertheless, in the particular case of leishmaniasis, intra-species transmission of *L. infantum* must be accounted for, since vertical transmission has been found to be possible in both humans [26] and dogs [27]. Furthermore, in the context of sylvatic transmission, other infecting modes should be considered beyond direct inoculation through the bite of an infected female sandfly. Recently, experimental evidence that *Leishmania* infection can occur by means of oral and intragastric transmission has been produced [28]. Based on this premise, since hares and rabbits (the major prey sources of the Iberian lynx [18]) have been found infected with *L. infantum* [29,30,31], predator–prey interaction should be considered as a possible route of infection for lynxes.

A positive association has been found between the presence of antibodies against FIV and *L. infantum* in cats. Nevertheless, the lack of studies prevents a correlation between the severity of FeL presentation and the FIV stage [32]. Even so, there are clinical reports on feline *L. infantum* co-infection with immune-suppressing viruses (e.g., FIV) and the development of severe clinical manifestations of FeL [33,34]. Since Iberian lynxes have been found infected with immunosuppressing conditions such as *Mycobacterium* spp., FIV, FeLV, and FCoV [19,21,22], such co-morbidities may be considered predisposing factors for the development of leishmaniasis. Moreover, due to this species’ reduced genetic variability due to the founding population effect, population decline, and conditioned breeding, any increased susceptibility to infection can be catastrophic [35]. Thus, it is of paramount importance to establish the tools to evaluate *Leishmania* exposure/infection in these animals. This can be addressed not only by the direct detection of the parasites but also through the evaluation of the animals’ humoral response. Therefore, in the context of the conservation of this endangered species, we explored for the first time the use of serological and molecular tools for screening *L. infantum* infection among reintroduced and wild-born lynxes living in a natural habitat. To approach our question, we took advantage of a group of samples previously collected from free-ranging Iberian lynxes living in the Portuguese Guadiana River Valley Park that have been monitored under the continuous and ongoing surveillance programs implemented by the Portuguese Institute for the Conservation of Nature and Forests (ICNF). Then, we evaluated the applicability of molecular and serological assays, which are routinely implemented for the identification of *Leishmania*, to perform both direct and indirect detection of these animals’ exposure to this protozoan parasite. A PCR-targeting leishmanial kinetoplast DNA was used for parasite screening in 35 DNA samples extracted from peripheral blood, with further confirmation of the infective species by internal transcribed spacer (ITS)–1 analysis. Furthermore, considering that there is no available gold-standard technique for anti-*Leishmania* antibody detection in cats and other felines, detection of anti-*Leishmania* antibodies was attempted from the plasma of 36 animals by means of indirect fluorescence antibody test (IFAT) and an enzyme-linked immunosorbent assay (ELISA) based on multiple *Leishmania*-specific, SPLA, rK39, and CPX, and the non-specific antigen, SECA. Since the role of feline humoral response to *Leishmania* infection is yet to be understood, the interpretation of anti-*Leishmania* antibody production profiles is essential to validate their use in epidemiological settings. 

## 2. Materials and Methods

### 2.1. Study Site and Sampled Population

Convenience sampling was carried out between 2018 and 2020 from 41 Iberian lynxes living in the Guadiana Valley Park (37°41′54″ N/7°38′51″ W), in the southeast of the Alentejo region, southern Portugal (Appendix A). Animals were captured from the wild, as part of the surveillance and monitoring program implemented by ICNF. The captures were performed within the framework of the EU LIFE+ Project LIFE10NA-ES/570). For the purpose of health monitorization and infectious disease surveillance, a total of 2 mL of whole blood was collected from each animal into an ethylenediamine tetra acetic acid (EDTA) tube for further analysis. For this study, ICNF and CEDVI made available a portion of the plasma and DNA.

### 2.2. Plasma and DNA Samples

From a total of 41 lynxes’ collected blood samples, plasma and DNA samples were available from 36 and 35 animals, respectively.

Plasma samples were separated after blood centrifugation, and aliquots were frozen at −20 °C until further processing. 

Total genomic DNA was extracted from 200 µL of EDTA blood using a commercial extraction kit (DNAeasy Blood and Tissue Kit, Qiagen, Hilden, Germany) according to the manufacturer’s instructions. Individual DNA aliquots were frozen at −20 °C until further processing. 

It was possible to obtain plasma and DNA from the same animal in 27 cases.

### 2.3. Serologic Assays

#### 2.3.1. Enzyme-Linked Immunosorbent Assay (ELISA)

Anti-*Leishmania* antibody detection was performed by ELISA through the adjustment of a protocol previously optimized for dogs [36,37,38]. Briefly, the ELISA was based on plate sensibilization with three *Leishmania*-specific antigens (SPLA, soluble *L. infantum* promastigotes antigens; rK39, recombinant k39 protein; *Lic*TXNPx, recombinant cytosolic peroxiredoxin protein, from now on referred to as CPX) and one antigen unrelated to *Leishmania* (SECA, soluble *Escherichia coli* antigens). SPLA was prepared from *L. infantum* strain MHOM/MA/67/ITMAP-263, as described by Santarém et al. [36]; rK39 antigen was obtained from Dr. Steven Reed (Infectious Disease Research Institute, Seattle, WA, USA); CPX was purified by means of affinity chromatography on a Ni-NTA column (Qiagen, Hilden, Germany) quantified by detergent-compatible (DC) protein assay (BioRad, Hercules, CA, USA), as previously described [37]. SECA was prepared at the PDG quantified by detergent compatible (DC) protein assay (BioRad, Hercules, CA, USA) and stored at −80 °C in single aliquots, as described by Lima et al. [38].

Ninety-six-well flat-bottom high binding microtiter plates (Greiner Bio-One, Frickenhausen, Germany) were coated with the four different antigens. Each antigen was diluted in 0.1 M carbonate buffer, pH 9.6, and 50 µL of each dilution containing 10 µg mL^−1^ SPLA, 1 μg mL^−1^ of rK39, 5 μg mL^−1^ CPX, and 10 µg mL^−1^ of SECA was added to the designated wells. Coated plates were incubated overnight (O/N) at 4 °C and blocked with 200 µL of PBS low-fat milk (3%) at 37 °C for 30 min. Plates were then washed three times with PBS-Tween (PBS-T) 0.05%. Serum samples were diluted at 1:800 in PBS-Tween 0.05% and dispensed in technical triplicates for each antigen (100 µL/well). After 30 min incubation at 37 °C, plates were washed three times with PBS-T 0.05%. Afterward, 100 µL of rabbit-produced anti-cat IgG antibody conjugated with horseradish peroxidase (Sigma, St. Louis, MO, USA) diluted 1:6000 was added to each plate. Plates were incubated for 30 min at 37 °C, and a wash step was repeated after incubation. To produce a readable product, the chromogenic substrate o-phenylenediamine dihydrochloride (OPD, Sigma, St. Louis, MO, USA) was diluted to 0.5 mg mL^−1^, activated by H_2_O_2_, and dispensed at 100 µL/well. After 10 min incubation in the dark, the reactions were stopped with 50 µL/well of HCl (3 M). Plates were read at *A*_492nm_ in an automatic ELISA plate reader (Synergy 2, BioTek Instruments, Winooski, VT, USA), and optical density (OD) values were recorded for every single measurement. Sera from a cat naturally infected with *L. infantum* was used as a positive control. All samples were tested in three technical triplicates for each antigen and repeated in at least two independent comparable assays. Optical densities were normalized to the blank, and the average result of two independent and comparable triplicates was calculated. The analysis of the serological response to *Leishmania*-specific antigens was performed with repeated-measures one-way ANOVA, with Friedman test, with Dunn’s multiple comparison test, and with Spearman’s correlation, using Prism 9 for Windows, version 9.4.0 (GraphPad software, San Diego, CA, USA). 

Calculation of seropositivity cut-offs was performed using the average seroreactivity of the 75% percentile plus 2× standard deviation. Graphical representation of seropositivity was depicted by the logarithm of the cut-off normalized OD values. 

#### 2.3.2. Indirect Fluorescence Antibody Test (IFAT)

The IFAT was performed following the protocol validated by Iatta et al. [39] with minor modifications. Briefly, stationary metacyclic *L. infantum* promastigote forms (strain MHOM/MA/67/ITMAP-263) were retrieved from a 7-day RPMI culture to be used as whole-parasite antigens fixed by cold acetone onto 12 multi-spot slides (MP Biomedicals, Eschwege, Germany). After three washing cycles with PBS, followed by centrifugation (1800× *g*, 10 min, at RT), promastigotes were resuspended in PBS to a final concentration of 5 × 10^6^ mL^−1^. Then, 20 µL of the suspension was distributed in each well. Slides were allowed to dry at 37 °C for 30 min and later fixed by acetone at −20 °C for 2 h. Afterward, slides were air-dried and preserved at −20 °C until use. 

Positive and negative controls were included in each slide, together with a blank. Sera from a feline leishmaniasis (FeL) case (confirmed by rK39 ELISA and parasite isolation through blood culture in Schneider medium) was used as the positive control and sera from a healthy, non-infected, indoor cat (rK39 ELISA and blood PCR negative) was used as a negative control. Each sample was diluted 1:20, 1:40, and 1:80 in PBS and 20 µL of each was placed in the center of the designated well. The slides were incubated for 30 min at 37 °C inside a moist chamber and rinsed with 0.001% PBS-Tween 80 for 2 min, followed by two washing steps of 5 min each in 0.001% PBS-Tween 80, and rinsed in distilled water for 2 min. After washing, slides were allowed to dry for 10 min at 37 °C and 20 µL of goat anti-cat IgG-fluorescein isothiocyanate conjugate (Sigma-Aldrich, Saint Louis, MO, USA) diluted 1:40 in 0.05% Evans blue were added to each well. The slides were again incubated inside a moist chamber at 37 °C for 30 min and then washed as described above. After the second washing procedure, slides were allowed to dry in the dark at 37 °C for 10 min. Ten microliters of mounting medium (Dako Fluorescense Mounting Media, Agilent, Santa Clara, CA, USA) were placed on every well before applying a coverslip. The slides were examined under a fluorescence microscope (Nikon Eclispe E400; Feasterville, PA, USA) at ×200 and ×400 magnifications. Images were captured by a digital camera (Nikon Digital sight DS-U1) and processed by a software (NIS-Elements F 3.2 software; Nikon, Tokyo, Japan). Fluorescence reading was normalized to the blank and negative controls. 

### 2.4. Polymerase Chain Reaction (PCR)

An initial screening for *Leishmania* was performed using the genus-specific PCR primers, RV1 (5′-CTTTTCTGGTCCCGCGGGTAGG-3′) and RV2 (5′-CCACCTGGCCTATTTTACACCA-3′), amplifying a 145 bp fragment of the leishmanial minicircle kinetoplast DNA (kDNA) [40]. The PCR reaction mix (final volume 25 µL) contained 2.5 µL of PCR 10× buffer; 0.5 µL of dNTP mix (10 mM of each dNTP; dNTP NZYmix, NZYTech, Portugal), 0.75 µL of 50 mM MgCl2, 0.8 µL of RV1 primer (10 mM), 0.8 µL of RV2 primer (10 mM), 5 µL of DNA template, and 0.1 µL of Taq DNA polymerase (Invitrogen Taq DNA Polymerase Recombinant, 5 U/µL, Carlsbad, CA, USA). Thermal cycle conditions were 94 °C for 3 min for initial denaturation, followed by 40 cycles of denaturation at 94 °C for 45 s, annealing at 59 °C for 50 s, and extension at 72 °C for 50 s. The final elongation step was carried out at 72 °C for 10 min. PCR products were submitted to electrophoresis for 60 min at 100 V on an ultrapure agarose gel (1.5%) prepared in 1× Tris-Acetate-EDTA (TAE) buffer supplemented with 0.2 µg/mL of nucleic acid stain (GreenSafe Premium, NZYtech, Lisboa, Portugal). Gel visualization was performed under ultraviolet light (530 nm) (BioRad Transilluminator, Hercules, CA, USA). A DNA sample extracted from cultured *L. infantum* promastigotes was used as a positive control. Contaminations were assessed by lack of amplification on a blank sample consisting of water instead of DNA. Positive samples were submitted to a second PCR reaction targeting *Leishmania* genus-specific ITS-1 region using primers LiTSR (forward) (5′-CTGGATCATTTTCCGATG-3′) and L5.8S (reverse) (5′-TGATACCACTTATCGCACTT-3′). The reaction occurred in a 25 µL final volume of reaction mix containing 2.5 µL of PCR buffer with MgCl, 0.5 µL of dNTP mix (10 mM of each dNTP; dNTP NZYmix, NZYTech, Portugal), 2.5 µL LiTSR primer (10 mM), 2.5 primer µL of L5.8S primer (10 mM), 5 µL of DNA template, and 0.1 µL of High Fidelity Taq DNA polymerase (Roche Expand High Fidelity System). Thermal cycle conditions were 94 °C for 3 min for initial denaturation, followed by 35 cycles of denaturation at 94 °C for 15 s; annealing at 53 °C for 30 s, and extension at 72 °C for 50 s. The final elongation step occurred at 72 °C for 7 min. The PCR products were analyzed on a 1.5% agarose gel stained with 0.2 µg/mL of nucleic acid stain (Greensafe Premium, NZYTech) and visualized on a transilluminator (ChemiDoc XRS+ system, Bio-Rad). Results were compared to a 100 bp molecular marker (Ladder V, NZYTech). The positive internally transcribed spacer (ITS)-1 PCR products were Sanger-sequenced in both directions using the same forward and reverse primers (AB Applied Biosystems, Thermo Fisher Scientific, Waltham, MA, USA). Nucleotide sequences were edited, analyzed, and compared with available sequences in the GenBank database, using the Basic Local Alignment Search Tool (BLAST; http://blast.ncbi.nlm.nih.gov/Blast.cgi (accessed on 26 October 2022). Alignment was performed with Clustal Omega from EBI web services [41]. The EMBL accession number for the ITS sequences used were AJ000289 (MHOM/TN/80/IPT1), AJ634344 (MHOM/PT/00/IMT260), AJ634343 (MHOM/ES/86/BCN16), AJ000294 (MHOM/CN/00/Wangjie1), AJ634361 (MHOM/SD/62/3S), AJ634370 (MHOM/SD/97/LEM3472). 

## 3. Results

Out of the 41 lynxes, 36 were tested against *Leishmania* antigens by IFAT and ELISA, and 35 were tested by PCR. Among these, 27 samples were analyzed by PCR, IFAT, and ELISA.

Eight DNA samples (8/35; 22.9%) tested positive using kDNA-PCR (a representative image of *Leishmania* spp. minicircle kDNA detection by PCR is depicted in Appendix A). Confirmation of *L. infantum* as an infective species was possible by nucleotide sequencing of the ITS-1 region in two samples. Sequences were identical among themselves and similar to other *L. infantum* ITS-1 sequences (Appendix A). Regarding the serological investigation, four samples were found to be IFAT-positive at the highest dilution of 1:40 (4/36) (Appendix A). The analysis of *Leishmania*-specific serological response was further addressed by ELISA, determining seroreactivity to three *Leishmania*-specific antigens (SPLA, rK39 and CPX) and SECA, as a non-specific antigen (Figure 1a; Appendix A). The average reactivity to SECA was significantly higher than the average seroreactivity to *Leishmania*-specific antigens (*p* < 0.0022), as opposed to the seroreactivity for all *Leishmania*-specific antigens (*p* > 0.999) (Figure 1a). 

To address *Leishmania*’s antigen-specific serological response, Spearman’s correlation was calculated from the seroreactivity obtained between the different antigens (Figure 1b). The highest correlations were observed for SPLA and rK39 (r = 0.88), followed by the correlation between rK39 and CPX (r = 0.87) and SPLA and CPX (r = 0.73). Conversely, the lowest correlations were observed for the pairing between *Leishmania*-specific antigens and SECA (0.49 > r < 0.64).

A stronger correlation between *Leishmania*-specific antigens suggested the presence of anti-*Leishmania* antibodies in the lynxes. To explore this possibility, seroreactivity cut-offs were established for the *Leishmania*-specific antigens and applied to the lynxes’ cohort (Figure 2).

Positivity to each *Leishmania*-specific antigen defined by the calculated ELISA cut-offs (Figure 2, Table 1) identified nine SPLA seropositive samples (9/36; 25.0%), ten rK39 seropositive samples (10/36; 27.8%), and eight CPX seropositive samples (8/36; 22.2%). Eight of the thirty-six studied samples (22.2%) were seropositive either to all *Leishmania*-specific antigens or to just two antigens. Agreement between IFAT and ELISA results (Table 1) was found for one IFAT-SPLA-positive sample (1/36; 2.78%), one IFAT-rK39-positive sample (1/36; 2.78%), and one IFAT-CPX-positive sample (1/36; 2.78%). Three IFAT-positive samples were negative on all ELISA antigens. Regarding agreement between serological and molecular assays, three PCR-positive samples (3/8) were positive for at least one *Leishmania*-specific ELISA antigen. Two PCR-positive samples (2/8) were positive for all *Leishmania*-specific antigens, while the sample exclusively positive for rK39 was PCR-positive. Agreement between PCR, IFAT and all-ELISA antigens was found for one sample. 

To further evaluate the performance of these cut-offs, the correlation in seropositive and seronegative groups was evaluated (Figure 3 and Appendix A). 

Among seropositive samples, all *Leishmania*-specific antigens evidenced a very strong and significant correlation, higher for rK39 with SPLA (r = 0.97), followed by rK39 with CPX (r = 0.85) and then CPX with SPLA (r = 0.80). Again, the specificity of these immunological interactions was supported by a weak and non-significant correlation between these antigens with SECA (below 0.45) (Figure 3a and Appendix A). The seronegative samples present decreased and less significant correlations. In this group, samples’ correlation between SECA and the three *Leishmania*-specific antigens was again not significant and was reduced (0.30 > r > 0.03) (Figure 3b and Appendix A). CPX presents a decreased non-significant correlation with SPLA (r = 0.35) (Figure 3b and Appendix A). The fact that SPLA and rK39 still presented significant correlations (r = 0.71) in the seronegative samples suggested that there was still some possibility of *Leishmania*-specific seroreactivity in this group. To address this, seronegative samples were stratified into two categories: high seronegative (sample’s ODs bellow 1× and 0.5× the cut-off) and low seronegative (0.5× below the cut-off) (Figure 3c,d). When considering correlations between the antigenic response in the high seronegative sub-group, a strong and significant association was found between rK39 and SPLA remains (r = 0.72), together with CPX and rK39 (r = 0.64) (Figure 3a and Appendix A). Considering the low seronegative, all correlations become even weaker and non-significant (Figure 3a and Appendix A).

Importantly, if considering the seroreactivity expressed by lynxes included in the high seronegative group (Appendix A), the degree of agreement between ELISA, IFAT, and PCR results increases significantly (Table 2). Agreement between the four IFAT-positive samples and those seroreactive to the three *Leishmania*-specific ELISA antigens increases to 75% (3/4) and reaches 100% for IFAT and SPLA-ELISA. Furthermore, among the eight PCR-positive samples, six (75%) match with any ELISA seropositive sample for this category. 

## 4. Discussion

Herein, we report the first survey of *L. infantum* infection involving 41 animals from a population of reintroduced and wild-borne Iberian lynxes living in the Portuguese Guadiana Valley Park. Seroreactivity against *L. infantum* was addressed by means of IFAT and an ELISA protocol based on three different *Leishmania*-specific antigens, including SPLA, rK39, and CPX, together with a *Leishmania* non-related antigen, SECA. Seroreactivity and seropositivity were compared to molecular detection of *Leishmania* by means of PCR. The quantitative antibody detection tools described here have been largely applied to CanL and FeL diagnosis and epidemiological surveys. The advantages of ELISA over IFAT include the possibility to detect specific anti-*Leishmania* antibodies by means of recombinant proteins and simultaneously compare seroreactivity against different antigens [36,37,38] while overcoming analysis bias related to IFAT interpretation. 

For screening *Leishmania* infection in the study group described, the approach based on kDNA amplification was followed as previously implemented in molecular surveys among *Leishmania* sylvatic hosts [42,43]. Previous studies on the prevalence of *L. infantum* infection among wild Iberian lynxes are reduced to one report from central Spain. In this study, Sobrino et al. [42] identified *L. infantum* DNA in one out of four animals (25%). In the same report, restriction fragment length polymorphism (PCR-RFLP) analysis of the infecting *Leishmania* strain was unrelated to the ones circulating among canids from the same region but similar to the pattern found on an *L. infantum* strain infecting a fox from a distinct geographical point. This highlights the existence of circulating *Leishmania* strains in sylvatic and domestic environments and the importance of typing these strains to better understand the ecology of the parasite. We found 22.9% (8/35) of lynxes to be PCR-positive (Table 1; Appendix A). Furthermore, *L. infantum* was confirmed as the infecting agent by PCR targeting ITS-1 region in two kDNA-PCR-positive samples (Appendix A). Due to DNA degradation and the small volume available for repetitions, ITS-1 amplification was not possible in all kDNA PCR-positive samples. The ITS-1 sequence was consistent with type A/B ITS sequence, that is, most prevalent in *L. infantum* (syn. *L. chagasi*) strains from the Mediterranean region, China, and South America [44]. 

Regarding IFAT, four samples were found to be IFAT-positive at a 1:40 dilution, supporting exposure to *Leishmania* but not leishmaniasis, if considering a cut-off of 80 for diagnosis [39].

A strong and relevant correlation was observed between seroreactivities against the different *Leishmania*-specific antigens among high seropositive lynxes, in contrast to a weaker or negative correlation observed between *Leishmania*-specific antigens and the non-*Leishmania*-related antigen (SECA) (Figure 3a and Appendix A). This supports the notion that lynxes develop specific antibodies against these parasites. The proposed seropositivity cut-offs (Figure 2) revealed the possibility to subcategorize this group of animals according to the strength of correlation between seroreactivities to the different *Leishmania*-specific antigens (Figure 3 and Appendix A). Among the high seropositive samples, defined by samples above the calculated cut-off, those animals presenting the highest correlation between *Leishmania*-specific antigens were clustered (Figure 3, Appendix A).

The performance of the three *Leishmania*-specific antigens used was distinct. Soluble *Leishmania* proteins (e.g., SPLA) have been widely applied as antigens on ELISA-based serological surveys for FeL [45,46,47,48,49], unlike the *Leishmania*-specific recombinant proteins used here. In a cut-off-independent approach, SPLA-based ELISA showed higher specificity than IFAT in detecting antibodies against *Leishmania* in cats (0.98 vs. 0.97) and a lower sensitivity (0.70 vs. 0.75) [48]. In our cut-off-based approach, SPLA-ELISA evidenced higher sensitivity than IFAT in detecting anti-*Leishmania* antibodies (9/36 vs. 4/36 animals) (Table 1). Among the IFAT-positive samples, only one was SPLA-ELISA and PCR-positive. On the other hand, among the nine SPLA seropositive samples, two were found to be PCR-positive (7.41%) (Table 1). 

*Leishmania* k39 protein, a 39-amino-acid repeat of a kinesin-related protein, is highly conserved, and its recombinant form (rK39) has shown improved sensitivity in detecting anti-*Leishmania* antibodies in dogs evidencing clinical manifestations of leishmaniasis. In a previous study on the applicability of three different *Leishmania* antigens in detecting antibodies in cats, the agreement between SPLA and rK39 seroreactivity was found moderate (k = 0.50) to weak [46], with SPLA showing higher sensitivity than rK39 [46,47] in detecting anti-*Leishmania* antibodies. Although it would be expected to find a larger number of SPLA seropositive samples, since this antigen presents a greater variety of *Leishmania* epitopes, in the present study, ten samples were rK39-positive (10/36) and nine samples were SPLA-positive (9/36) (Table 1), with eight samples presenting agreement between individual results. Broadly, these results reflect a similar performance of these two antigens in detecting anti-*Leishmania* antibodies in lynxes. The differences observed when comparing the performance of rK39 and SPLA in our cohort with the available studies in cats from Brazil [46] might be related to the distinct ecological background of the animals, which lead to distinct patterns of seroreactivity, and also to the calculation of a diagnostic cut-off based on samples obtained from cats with negative *Leishmania* spp. cytological examination of lymph nodes and bone marrow aspirates. This will influence the cut-offs. Animals without cytological evidence of infection were used to determine the cut-offs. Due to being more prone to cross-reactivity, SPLA is more sensitive, although less specific [50]. Thus, rK39 would have fewer positive animals. Due to the limitations of our study, the same population was used to define the cut-offs and to search for seroreactive samples. Although the use of SECA strongly suggests that cut-off-associated seroreactivity is *Leishmania*-specific, it is a distinct approach to what was used for the above-mentioned cat studies. Our seropositivity represents abnormal seroreactivity in the population, and our data suggest that this parameter is not distinct between SPLA and rK39.

The identification of cryptic *Leishmania* infections remains a diagnostic pitfall, with various possible diagnostic presentations including negative serology (confirmed by different serological methods including IFAT, ELISA, and direct agglutination test, DAT) and positive PCR [7]. Our findings support similar presentations, with three of the eight lynxes (37.5%) presenting a positive PCR and negative serology (Table 1). 

In some cases, *L. infantum* cytosolic peroxiredoxin, described herein as CPX, has proved to be a sensitive and specific antigen for the detection of anti-*Leishmania*-specific antibodies among sub-clinically infected dogs and children, when compared to rK39 [36,37]. The performance of CPX in detecting anti-*Leishmania* antibodies in felines has never been reported; therefore, its capacity to recognize subclinical infections is unknown. In our findings, CPX seroreactivity was found to be *Leishmania*-specific as it is strongly correlated with SPLA and rK39 among high-seropositive animals. The correlation of CPX or rK39 with clinical presentations was not possible in the current context, since data on the health background of these animals were not made available for the purpose of this particular study. Worldwide, FeL is a rare clinical presentation, and seroreactivity against *Leishmania* antigens and direct parasitological identification can be found in sub-clinically infected cats [25]. Within the Iberian Peninsula, the prevalence of *Leishmania* infection (reported based on positive blood PCR) is often found to be higher than seropositivity [45]. In our study, the prevalence of infection determined by PCR (identification of leishmanial kinetoplast DNA) was 22.9% (8/27). For serological assays, the applied ELISA cut-off determined seropositivity for all *Leishmania*-specific antigens in 22.2% (8/36) of the samples, while IFAT identified 11.1% of positive samples at 1:40 dilution (4/36). Agreement between IFAT and ELISA (considering all *Leishmania* antigens) was found in only 2.8% of the samples (1/36). Agreement between IFAT and PCR was found in 11.1% of the samples (3/27) and positive IFAT results agreed with all ELISA-*Leishmania* specific antigens and PCR in 3.7% of the studied samples (1/27). The discrepancy between results among the different techniques can be justified by technical aspects, such as small sample size, the cut-offs applied on ELISA antigens, and the PCR performance. The discrepancy attributed between direct and indirect diagnostic test results can be justified, among other aspects, by (1) the timeframe required to seroconvert; (2) the possibility of some of the infected animals never having undergone seroconversion, as in dogs; (3) the possibility that seropositive animals have a false-negative blood PCR result. Regarding PCR, previous data support the notion that bone marrow, lymph nodes, and spleen are tissues harboring a higher parasite burden and therefore should be elected for PCR analysis over peripheral blood, which offers lower sensitivity [51,52]. However, sampling of such tissues is invasive and not suitable to be performed in field conditions and represents a limitation to epidemiological surveys. Furthermore, invasive sampling methods are inadequate or against the scope of the conservation program for the Iberian lynx. The discrepancy of PCR results between kDNA and ITS-1 primers can be justified by the fact that the amplification of *Leishmania* kinetoplast DNA is often superior, as the kinetoplast minicircle contains over 10,000 copies per cell. This biological aspect of *Leishmania’s* DNA organization makes kDNA targets good candidates for *Leishmania* screening. On the other hand, ITS is located between the small subunit and the large subunit or ribosomal RNA genes and is therefore present in lesser abundance. Different detection thresholds have been reported for the mentioned *Leishmania* PCR targets, depending on the protocol and analyzed tissues. Overall, ITS-1 targets present a lower detection threshold compared to kDNA primers [53]. 

Considering the data generated, *Leishmania* surveillance methods in Iberian lynxes should include blood collection to perform PCR for infection detection, in conjunction with serological assessment. Moreover, due to the lack of evidence-based capacity of serological tests to discriminate infection, we think that multiple antigens are the best answer for a quicker understanding of serological assessment in these felines. The minimal panel that would be ideal should combine a potentially less specific but more sensitive antigen such as SPLA in conjunction with a potentially more specific but less sensitive recombinant antigen such as rK39; in addition, SECA should be used for a better distinction between *Leishmania*-specific seroreactivity and potentially unknown cross-reactivity.

## 5. Conclusions

Herein, we report the first survey of *L. infantum* infection among 41 animals from a population of reintroduced and wild-borne Iberian lynxes living in the Portuguese Guadiana Valley Park. Our findings, based on a serosurvey performed by ELISA and IFAT, supported by molecular identification of *Leishmania infantum*, evidence that this endangered species is susceptible to infection with *L. infantum*. The lack of previous scientific evidence supporting this topic, aggravated by the scarcity of research and knowledge of felines’ immune response to *Leishmania*, makes it impossible to compare these results with previous data. However, as previously described for dogs, we evidenced that there is an improved sensitivity and specificity of the serological approach to *Leishmania* exposure when using a triple combination of *Leishmania*-specific antigens, namely SPLA, rK39, and CPX, combined with a non-related antigen, which improves the assay’s sensitivity and specificity, allowing a better understanding of *Leishmania* infection dynamics among this endangered species. Therefore, it should be considered in future studies. For the proper assessment of the impact of *L. infantum* infection on the Iberian lynxes’ health and conservation, future surveillance programs should include the longitudinal evaluation of these animals, considering medical physical examination with clinical laboratory analyses, serological and molecular screening, and integration of findings regarding *Leishmania* infection in the context of co-infections with other infectious diseases of vectorial and non-vectorial transmission. A full validation of the biological relevance of the data presented herein requires the follow-up on the seropositive suspect animals to assess the prognosis for both clinical and parasitological conditions. 

## Figures and Tables

**Figure 1 microorganisms-10-02447-f001:**
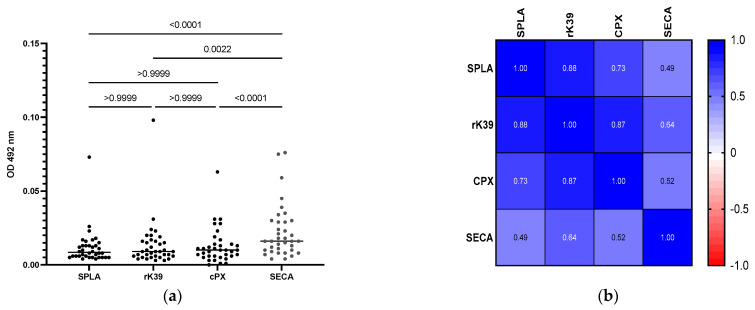
Iberian lynxes’ (n = 36) seroreactivity to SPLA (soluble *L. infantum* promastigotes antigens); rK39 (recombinant k39 protein); CPX (recombinant cytosolic peroxiredoxin protein) and SECA (soluble *Escherichia coli* antigens). (**a**) Graphical representation of optical densities (ODs) at 492 nm for each antigen. Each dot represents the average OD results, of at least two independent assays performed in triplicate, for each sample. The horizontal lines indicate the average OD result for each antigen. Statistical analysis was performed using RM one-way ANOVA, with Friedman test with Dunn’s multiple comparison test. Individual *p*-values are depicted, associated with each comparison; (**b**) non-parametric Spearman’s correlation between *Leishmania*-specific antigens (SPLA, rK39, CPX, and SECA). Strong correlation (r > 0.7) between all *Leishmania* antigens vs. low correlation between *Leishmania*-specific antigens and SECA (r < 0.64). Very strong correlation is considered for 0.9 > r < 1; strong correlation is considered for 0.7 > r < 0.89; moderate correlation is considered for 0.4 > r < 0.69; weak correlation is considered for 0.4 > r < 0.69; and very weak correlation is considered for r < 0.19.

**Figure 2 microorganisms-10-02447-f002:**
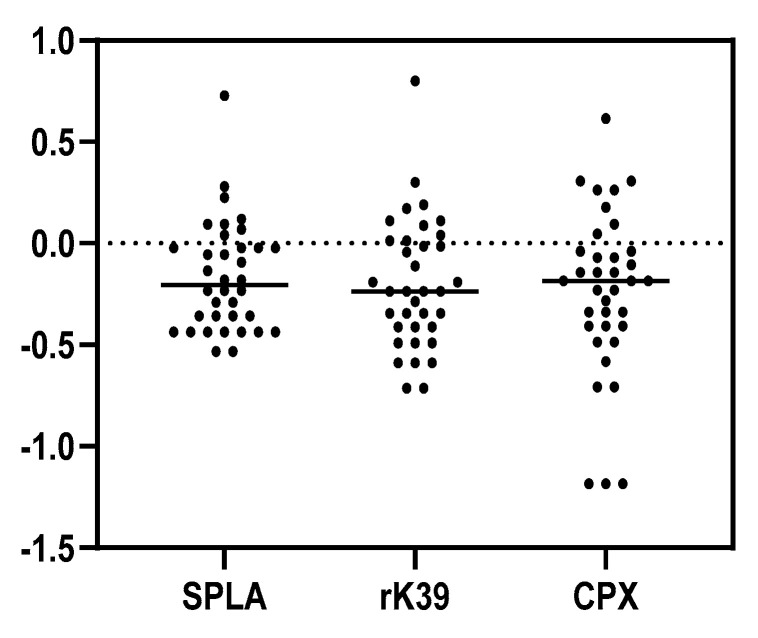
Graphical representation of Iberian lynxes’ (n = 36) seropositivity to SPLA, rK39, and CPX. Each dot represents the logarithm (log10) of the average OD results of at least two independent assays, performed in triplicate for each sample, normalized to the cut-off for each antigen. The population cut-off was determined based on the average ODs for the 75% population quartile plus 2 SD. The horizontal black line represents the average result for each antigen.

**Figure 3 microorganisms-10-02447-f003:**
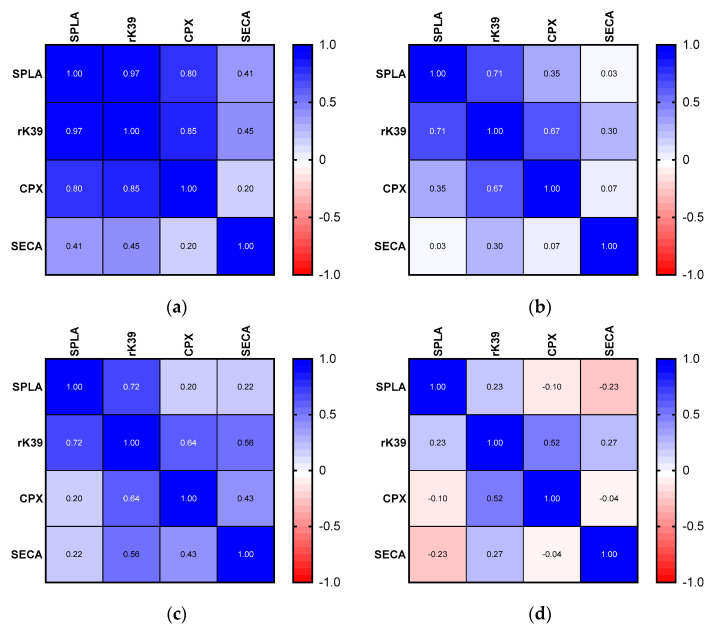
Spearman’s correlation for seroreactivity to SPLA, rK39, CPX, and SECA within stratified groups of seropositive and seronegative lynxes. (**a**) Spearman’s correlation for all seropositive samples; (**b**) Spearman’s correlation for all seronegative samples; (**c**) Spearman’s correlation for seronegative samples with seroreactivity values between a 0.5 and 1 cut-off (high seronegative); (**d**) Spearman’s correlation for all low seronegative samples with seroreactivity values lower than a 0.5 cut-off (low seronegative). A very strong correlation is considered for 0.9 > r < 1; strong correlation is considered for 0.7 > r < 0.89; moderate correlation is considered for 0.4 > r < 0.69; weak correlation is considered for 0.4 > r < 0.69; and very weak correlation is considered for r < 0.19.

**Table 1 microorganisms-10-02447-t001:** Positivity of *L. infantum* infection detected by ELISA *Leishmania*-specific antigens (SPLA, soluble *L. infantum* promastigotes antigens; rK39, recombinant k39 protein; CPX, recombinant cytosolic peroxiredoxin protein; SECA, soluble *Escherichia coli* antigens), IFAT, and blood PCR targeting kDNA (primers RV1 and RV2).

Diagnostic Tool	Total Analyzed *N*	Total Positive *n*	% Positive
SPLA	36	9	25.0
SPLA + rK39	36	8	22.2
SPLA + CPX	36	8	22.2
SPLA + rK39 + CPX	36	8	22.2
SPLA + IFAT	36	1	2.8
SPLA + rK39 + IFAT	36	1	2.8
SPLA + rK39 + CPX + IFAT	36	1	2.8
SPLA + PCR	27	2	7.4
SPLA + rK39 + PCR	27	2	7.4
SPLA + rK39 + CPX + PCR	27	2	7.4
SPLA + IFAT + PCR	27	1	3.7
SPLA + rK39 + IFAT + PCR	27	1	3.7
SPLA + rK39 + CPX + IFAT + PCR	27	1	3.7
rK39	36	10	27.8
rK39 + CPX	36	8	22.2
rK39 + IFAT	36	1	2.8
rK39 + CPX + IFAT	36	1	2.8
rK39 + PCR	27	2	7.4
rK39 + CPX + PCR	27	2	7.4
rK39 + IFAT + PCR	27	1	3.7
rK39 + CPX + IFAT + PCR	27	1	3.7
CPX	36	8	22.2
CPX + IFAT	36	1	2.8
CPX + PCR	27	2	7.4
CPX + IFAT + PCR	27	1	3.7
IFAT	36	4	11.1
PCR	35	8	22.9
IFAT + PCR	27	3	11.1

**Table 2 microorganisms-10-02447-t002:** Positive associations between high seronegative and seropositive samples are detected by ELISA *Leishmania* specific antigens (SPLA, rK39, CPX), IFAT, and blood PCR targeting kDNA (primers RV1 and RV2).

Associations between Diagnostic Tools	Total *n*	Overall %
ELISA antigens		
SPLA + rK39	18/36	50.0
SPLA + CPX	17/36	47.2
rK39 + CPX	18/36	50
SPLA + rK39 + CPX	16/36	44.4
IFAT and ELISA antigens		
IFAT + SPLA	4/36	11.11
IFAT + rK39	3/36	8.33
IFAT + CPX	3/36	8.33
IFAT + SPLA + rk39	3/36	8.33
IFAT + SPLA + CPX	3/36	8.33
IFAT + rK39 + CPX	3/36	8.33
IFAT + SPLA + rK39 + CPX	3/36	8.33
PCR and ELISA antigens		
SPLA + PCR	6/27	22.2
rK39 + PCR	6/27	22.2
CPX + PCR	6/27	22.2
SPLA + rK39 + PCR	6/27	22.2
SPLA + CPX + PCR	6/27	22.2
rK39 + CPX + PCR	6/27	22.2
SPLA + rK39 + CPX + PCR	6/27	22.2
PCR and IFAT		
IFAT + PCR	3/27	11.1
All diagnostic tools		
SPLA + rK39 + CPX + IFAT + PCR	3/27	11.1

## Data Availability

Not applicable.

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
