# Peer review of "Serological and Molecular Survey of Leishmania infantum in a Population of Iberian Lynxes (Lynx pardinus)"

_microorganisms, 2022, doi:10.3390/microorganisms10122447_

Round 1

Reviewer 1 Report

Overall, the article is well written and the results are interesting, with a thorough evaluation of the methods used for Leishmania diagnosis in lynxes. However, the Discussion should be improved in terms of overall structure and to include discussions of causes for the observations and differences to other studies - some instances are pointed out below, but the authors should carefully revise the entire Discussion. The Discussion should also include a proposal of an optimal method or algorithm for surveillance of Leishmania infection in lynxes

1. More information should be provided regarding the blood collection and ethics. Was the blood collected part of the surveillance and monitoring program or specifically for the project? If for the project, does it have specific ethics clearance for this collection? Please clarify in the manuscript.

2. line 307: "appeared to be" or "was"?

3. it would help the reader if Tables 2 were better organized. For example, on Table 1, the sequence could be: SPLA alone, SPLA + rK39, SPLA+rK39+CPX, SPLA+rK39+CPX+IFAT+PCR. Then, start at rK39 alone, rK39+CPX, etc. Preferably, with left alignment. Other options are acceptable, but the order of appearance should follow a regular order, which is not in the current version. It should be indicated on the Tables that SPLA, rK39 and CPX refer to ELISA

4. Lines 351-2: Clarify for the reader who hasn't read the M&M section, that ELISA using SECA wasn't Leishmania-specific. E.g. "... and ELISA with a non-related antigen, SECA"

5. Line 351: Also make it clear that the ELISA protocol used the Leishmania antigens separately, rather than "a multi Leishmania-specific antigen"

6. Lines 405-6: Please explain or correct. How is an agreement in 8 samples (out of 36?) a strong agreement? 

7. Table 1: As per the text (line 301), the table indicates agreement between the methods/antigens. The prevalence, as stated in the Table title, could be defined as one method positive, rather than both, and so it should be clear for the reader.

8. Lines 404-5: "and nine samples were SPLA positive (10/36)" - correct.

9. Lines 400-406: A comparison between the two studies should be made, with possible explanations for the differences.

10. Lines 410-1: please discuss possible explanations for positive PCR and negative serology.

11. Lines 421-34: What is the relevance to this work? The authors did not analyze co-infections here nor clinical signs of leishmaniasis in the lynxes. Please remove or indicate how it relates to the work.

12: Lines 435-45: Similarly, the present work did not evaluate sources of infection for the lynxes. Please remove or indicate how it relates to the work.

13: Lines 450-4: to be able to state this, the authors should have discussed and proposed the optimal diagnosis combination, based on their results. This thorough discussion of the relative merits of the diagnostic methods evaluated and a proposal of an optimal method or algorithm for surveillance of Leishmania infection in lynxes should be added to the Discussion, replacing the paragraphs mentioned above (points 11 and 12), which are not directly related to the work presented here. 

Author Response

To respond to the reviewer's concerns and advice, we submit in this document a detailed reply to the reviewer’s comments and have uploaded an updated version of the manuscript.

The uploaded manuscript versions with the appendix “Tracked Changes” in the file name contain all the changes made to the original document. They can be seen using the “Track Changes All Markup” feature of Microsoft Word. We also highlighted all modified sections in yellow. The version with the appendix “clean” corresponds to the current version of the manuscript with all modifications included and accepted.

Foremost, we would like to thank the reviewers for their reviews. To improve clarity, in all replies to the reviewers' concerns, we have applied a point-by-point response method. In this document, the original text by the reviewer (in bold) is followed by the author's reply/comment. Besides the authors comments, we also detected some minor mistakes (references formatting and text editing) that were also corrected.

Overall, the article is well written and the results are interesting, with a thorough evaluation of the methods used for Leishmania diagnosis in lynxes. However, the Discussion should be improved in terms of overall structure and to include discussions of causes for the observations and differences to other studies - some instances are pointed out below, but the authors should carefully revise the entire Discussion. The Discussion should also include a proposal of an optimal method or algorithm for surveillance of Leishmania infection in lynxes

Authors’ response: we thank Reviewer 1 for her/his positive evaluation of our manuscript. The Discussion section has now been revised, in order to also include a proposal of an optimal method for surveillance of Leishmania infection in lynxes.

Now between lines 505 and 513 now reads: “Considering the data generated, Leishmania surveillance methods in Iberian lynxes should include blood collection to perform PCR for infection detection, in conjunction with serological assessment. Moreover, due to the lack of evidence-based capacity of serological tests to discriminate infection, we think that multiple antigens are the best answer for a swifter understanding of serological assessment in these felines. The minimal panel that would be ideal should combine the potentially less specific antigen but more sensitive antigen like SPLA in conjunction with a potentially more specific but less sensitive recombinant antigen like rK39, also SECA should be used for a better distinction between Leishmania-specific seroreactivity and potentially unknown cross reactivity.”

  1. More information should be provided regarding the blood collection and ethics. Was the blood collected part of the surveillance and monitoring program or specifically for the project? If for the project, does it have specific ethics clearance for this collection? Please clarify in the manuscript.

Authors’ response:: as stated in the Materials and Methods section (2.1. Study site and sampled populations), between lines 151-153 of the revised manuscript: “Animals were captured from the wild, as part of the surveillance and monitoring program implemented by the Institute for the Conservation of Nature and Forests (ICNF). The captures were performed within the framework of the EU LIFE+ Project LIFE10NA-ES/570).”

We have also added a new statement, under Institutional Review Board Statement. On line 575 of the revised manuscript now reads: “All clinical procedures complied with the Portuguese legislation for the protection of animals used for scientific purposes (i.e. Decree-Law no. 113/2013, of 7 August 2013), which transposes European legislation (i.e. Directive 2010/63/EU of the European Parliament and of the Council, of 22 September 2010).”

An ethical approval has been institutionally waived in this case, as the procedures were those of veterinary clinical assistance to the animals. In fact, blood samples were collected for clinical analysis (haematological, serum/plasma biochemistry and screening of infectious diseases) and leftover sample material was used for the specific serological and molecular investigations presented in the manuscript. Although blood was not collected on purpose for this study, results were useful for the clinical characterization of the animals.

line 307: "appeared to be" or "was"?

Authors’ response: Reviewer 1 is right in her/his remark. Wording has now been updated from “appeared to be” to “was” in line 337 of the revised manuscript.

it would help the reader if Tables 2 were better organized. For example, on Table 1, the sequence could be: SPLA alone, SPLA + rK39, SPLA+rK39+CPX, SPLA+rK39+CPX+IFAT+PCR. Then, start at rK39 alone, rK39+CPX, etc. Preferably, with left alignment. Other options are acceptable, but the order of appearance should follow a regular order, which is not in the current version. It should be indicated on the Tables that SPLA, rK39 and CPX refer to ELISA

Authors’ response: We agree with the reviewer's recommendation that a regular order of the antigens would help readability. Thus, we have modified Tables 1 and 2 according to the recommendations.

Lines 351-2: Clarify for the reader who hasn't read the M&M section, that ELISA using SECA wasn't Leishmania-specific. E.g. "... and ELISA with a non-related antigen, SECA"

Authors’ response: As suggested by Reviewer 1 we modified the wording in this section. Lines 385-387 now read: “Seroreactivity against L. infantum was addressed by means of IFAT and an ELISA protocol based on three different Leishmania-specific antigens, including SPLA, rK39, CPX, together with a Leishmania non-related antigen, SECA.”

Line 351: Also make it clear that the ELISA protocol used the Leishmania antigens separately, rather than "a multi Leishmania-specific antigen"

Authors’ response: this has also been made clearer in the text section added to clarify the previous point, the wording multi Leishmania-specific antigen was removed.

Lines 405-6: Please explain or correct. How is an agreement in 8 samples (out of 36?) a strong agreement? 

Authors’ response: The text was inaccurate, there is no statistical basis was for the definition of the mentioned agreement as “strong”. Thus we toned down the sentence that now reads between lines 444 and 445 of the revised manuscript: “(…) in the present study, ten samples were rK39 positive (10/36) and nine samples were SPLA positive (9/36) (Table 1), with eight samples presenting agreement between individual results”.

Table 1: As per the text (line 301), the table indicates agreement between the methods/antigens. The prevalence, as stated in the Table title, could be defined as one method positive, rather than both, and so it should be clear for the reader.

Authors’ response: Reviewer 1 is right in her/his remark. We have adapted table title to read as “Table 1. Positivity for L. infantum infection detected by ELISA, IFAT and blood PCR targeting kDNA (primers RV1 and RV2).

Whereas positivity regards the positive results of each test, prevalence is the total number of positive results regardless of the test. Put another way, prevalence regards the number (and proportion) of lynxes that were positive in at least one test.

Lines 404-5: "and nine samples were SPLA positive (10/36)" - correct.

Authors’ response: text has been corrected to read as: “… and nine samples were SPLA positive (9/36)” (line 444 of the revised manuscript).

Lines 400-406: A comparison between the two studies should be made, with possible explanations for the differences

Authors’ response: To address the need to compare these studies this section was added in lines 446 and 458 of the revised manuscript.

“The differences observed when comparing the performance of rK39 and SPLA in our cohort with the available studies in cats from Brazil [46], might be related to the distinct ecological background of the animals, that lead to distinct patterns of seroreactivity, and also with the calculation of a diagnostic cut-off based on samples obtained from cats with negative Leishmania spp by cytological examination of lymph node and bone marrow aspirates. This will influence the cut-offs. Due to being more prone to cross-reactivity, SPLA is more sensitive although less specific [50]. Thus rK39 would have fewer positive animals. Due to the limitations of our study, the same population was used to define the cut-offs and to search for seroreactive samples. Although the use of SECA strongly suggests that cut-off associated sero-reactivity is Leishmania-specific, it is a distinct approach for what was done for the mentioned cat studies. Our seropositivity represents abnormal seroreactivity in the population and our data suggest that this parameter is not distinct between SPLA and rK39.”

Lines 410-1: please discuss possible explanations for positive PCR and negative serology.

Authors’ response: the most likely explanation for a positive PCR result and a negative serology is the fact that an animal might be at the so-called seroconversion period. In dogs, the seroconversion period lasts 3 months on average. That circumstance may have also been a possible explanation here. This has now been included in the text between lines 484 and 504.

Lines 421-34: What is the relevance to this work? The authors did not analyze co-infections here nor clinical signs of leishmaniasis in the lynxes. Please remove or indicate how it relates to the work.

Authors’ response: Reviewer 1 is right in her/his remark. Text from lines 424 to 434 has now been deleted.

The aim of this section is to highlight the existence of multiple Leishmania co-infections in Iberian cats, the closest relative to the Lynx in the Iberian Pensinsula, especially immunosuppressing pathogens. We agree that this section in here was out of context, and we moved most of this information to the introduction. This contributed to emphasise the risk of Leishmania infection for these felines.

Lines 435-45: Similarly, the present work did not evaluate sources of infection for the lynxes. Please remove or indicate how it relates to the work.

Authors’ response: The aim of this section was to highlight the different routes of infection with Leishmania for the Lynx in the Iberian Pensinsula. We agree that this section in here was out of context, and we moved most of this information to the introduction. This contributed to emphasise the risk of Leishmania infection for these felines.

Lines 450-4: to be able to state this, the authors should have discussed and proposed the optimal diagnosis combination, based on their results. This thorough discussion of the relative merits of the diagnostic methods evaluated and a proposal of an optimal method or algorithm for surveillance of Leishmania infection in lynxes should be added to the Discussion, replacing the paragraphs mentioned above (points 11 and 12), which are not directly related to the work presented here.

Authors’ response: The definition of a definitive diagnostic method would require the follow-up of the animals to address the fate of the observed distinct serological and molecular patterns. Still this was not the goal of this manuscript. We aimed at demonstrating the capacity of available serological approaches to be used in these animals in epidemiological surveys. We highlight in the end of the discussion the merit of a combination of results from different antigens to improve the sensitivity of detection.

Reviewer 2 Report

This works presents results of molecular and serological analyses addressing Leishmania infection in a population of Iberian lynxes. I would like to comment on aspects related to the hypothesis, methodology, and interpretation of results. I believe the manuscript can be improved in terms of clarity and specificity of the work conducted:

1) The authors statet that "... the impact of Leishmania infantum infection on the conservation of the endangered Iberian lynx (Lynx pardinus) is unknown." Is there a reason to think L. infantum infection could represent a risk to the Iberian lynx population? Are there any informations pointing to the impact of this parasite in the health of these animals? What is the working hypothesis in this study? What does prompt the analysis? Is it just exploratory or guided by any preliminary information? This should be stated in the text.

2) Positive serology is obtained at very high titres only, meaning these may not be actual positive result. The serological methods seem to be not validated previously in this population of animals, which is understandable, but results should be taken with caution. Only 1 in 27 samples returned positive results for all tests used (1 molecular, 2 serological), is this really enough to state lynxes are susceptible to Leishmania infection? Also, serological positive results just coincided in one sample? This seems to be actually a limitation: small sample size and discordant results. 

3) Sample. Is this a conveninece sampling taking advantage of the surveillance and monitoring program by ICNF? Or was sampling designed ad hoc for assessing prevalence of Leishmania infection in the lynx population? Please, clarify. 

4)  Please provide more details in how the sample was selected, the animal captured and handled during the process.

And, where 2 mL collected from these animals (endangered species) just to assess whether they were exposed or infected by Leishmania? Was this accepted by and ethics or animal welfare committee? Please develop this aspect.    

5) DNA used. DNA extraction started from 200 uL EDTA blood, please indicate elution volume and elution solution.

6) ELISA. Anti-cat IgG antibody is used, but not anti-lynx IgG. I understand the anti-lynx antibody may not be available, but I would take this as a reason for caution in the interepretation of the results. Is there any information available supporting the equivalence of anti-cat and anti-lynx IgG? If not, caution with the results.  

"Calculation of seropositivity cut-offs was performed using the average seroreactivity of the 75% percentile plus 2x standard deviation" Please, explain the rationale for this.

7) IFAT. Negative control: wasn't it possible to use sera from a lynx from an area non-endemic for leishmaniasis?

Same comment as above regarding anti-cat IgG. These are limitations that point towards caution regarding the interpretation of the results.  

8) Accession numbers DNA sequences. Please, explain the rationale for using the following sequences and no other available in GenBank, AJ634344 (MHOM/PT/00/IMT260), AJ634343 (MHOM/ES/86/BCN16), AJ000294 247 (MHOM/CN/00/Wangjie1), AJ634361 (MHOM/SD/62/3S), AJ634370 248 (MHOM/SD/97/LEM3472)

9) Did the authors submit the DNA sequences they generated to the GenBank? Is there a plan for doing this? THis is good practice when reporting results related to DNA sequences.

10) Discussion. Please refer results to the specific study sample, this cannot be generalized to the Iberian lynx population unless the study design address this specificially.

11) Discussion. "In the same report, restriction fragment length polymorphism (PCR-RFLP) analysis of the infecting Leishmania strain was unrelated to the ones circulating among canids from the same region, but similar to the pattern found on a L. infantum strain infecting a fox from a distinct geographical point." What is the purpose of this sentence? Is that the authors want to distinguish a sylvatic from a domestic cycle? Was that study designed to address that question?

12) Discussion. "Due to DNA degradation and small volume, ITS-1 amplification was not possible in all kDNA PCR positive samples." So DNA degradation affected the performance of ITS-1 PCR but not that of kDNA PCR? And what do the authors mean by small volume? 2 mL blood were collected from the animals, and 200 uL used for DNA extraction, meaning this could have been repeated 10 times allowing ample volume for PCR testing. Besides, samples were stored at -20ºC, meaning good storage practice to avoid DNA degradation. Could it be possible to get more details on the timings and process of sample collection and preparation for analysis to understand where degradation could have occurred?  

13) Discussion. "... no data is available on the health background of these animals." How is this possible? Specimens from an endangered species are captured to assess Leishmania infection and no data is registered regarding their health status? The animals were not explored for signs compatible to leishmaniasis? 

14) Discussion, line 433. Correct "co-mobilities"

15) " Recently, experimental evidence that Leishmania infection can occur by means of oral and intragastric transmission has been produced [52]. Based on this premise, since hares and rabbits (the major prey source of the Iberian lynx [31]) have been found infected with L. infantum [53–55], predator-prey interaction should be considered as a possible route of infection for lynxes." Is not more reasonable to consider that Leishmania infection would occur by the bites of infected sandflies? And in case this is true, will this be a transient or a persistent infection? Will the latter return more consistent results in terms of concordance between serological and molecular tests?

16) Conclusions "Our findings support evidence that this endangered species is susceptible to infection with L. infantum." This is a strong statement, maybe an important limitaton to indicate is the lack of concordance between tests, plus the inability of demonstrating the parasite in tissue.

17) Supplementary materials. Maps illustrating capture points would benefit of overlapping information on Leishmania presence, if available. Otherwise it should be explained whay those capture areas were selected if information on Leishmania presence is not available. 

Author Response

To respond to the reviewer's concerns and advice, we submit in this document a detailed reply to the reviewer’s comments and have uploaded an updated version of the manuscript.

The uploaded manuscript versions with the appendix “Tracked Changes” in the file name contain all the changes made to the original document. They can be seen using the “Track Changes All Markup” feature of Microsoft Word. We also highlighted all modified sections in yellow. The version with the appendix “clean” corresponds to the current version of the manuscript with all modifications included and accepted.

Foremost, we would like to thank the reviewers for their reviews. To improve clarity, in all replies to the reviewers' concerns, we have applied a point-by-point response method. In this document, the original text by the reviewer (in bold) is followed by the author's reply/comment. Besides the authors comments, we also detected some minor mistakes (references formatting and text editing) that were also corrected.

 This works presents results of molecular and serological analyses addressing Leishmania infection in a population of Iberian lynxes. I would like to comment on aspects related to the hypothesis, methodology, and interpretation of results. I believe the manuscript can be improved in terms of clarity and specificity of the work conducted:

Authors’ response: we are grateful for the comments made by Reviewer 2 about our work, which have given us the opportunity to improve the value of the manuscript.

The authors statet that "... the impact of Leishmania infantum infection on the conservation of the endangered Iberian lynx (Lynx pardinus) is unknown. " Is there a reason to think L. infantum infection could represent a risk to the Iberian lynx population? Are there any informations pointing to the impact of this parasite in the health of these animals? What is the working hypothesis in this study? What does prompt the analysis? Is it just exploratory or guided by any preliminary information? This should be stated in the text.

Authors’ response: We thank the reviewer for the attention on the relevance of the work. We will provide detailed query specific answers in this sections.

Is there a reason to think L. infantum infection could represent a risk to the Iberian lynx population?

Authors’ response: We think that from the available information on Leishmaniasis in felines that L. infantum infection can represent a risk to the Iberian lynx population. The Iberian lynxe’s infection by L. infantum has been reported only once by Sobrino et al. 2008 who found 1 infected lynx among 4 studied animals. Furthermore, L. infantum has also been found to infect other wild felines (Cardoso et al. 2021). We also describe in the introductory section, that there is enough evidence that domestic cats living in L. infantum endemic regions may become infected by L. infantum and even develop disease, that can be more severe in the presence of immunosuppressive pathogens like FIV.  Thus, it is likely that the same scenarios exist for Lynxs, considering their reduced genetic variability due to the founding population effect and conditioned breeding, any increased susceptibility to infection can be catastrophic.

Are there any informations pointing to the impact of this parasite in the health of these animals?

Authors’ response: The main body of work available is on domestic cats. They can become diseased. Most common clinical signs and clinicopathological abnormalities compatible with FeL include lymph node enlargement and skin lesions such as ulcerative, exfoliative, crusting or nodular dermatitis (mainly on the head or distal limbs), ocular lesions (mainly uveitis), feline chronic gingivostomatitis syndrome, mucocutaneous ulcerative or nodular lesions, hypergammaglobulinaemia and mild normocytic normochromic anaemia. Clinical illness is frequently associated with impaired immunocompetence, as in case of retroviral coinfections. Thus, the available information suggests a possible impact in the heath of Lynx’s.

What is the working hypothesis in this study?

Authors’ response: The working hypothesis is to gain knowledge on the frequency of Leishmania infection in a group of wild-borne Iberian lynxes living in the only reintroduction site in Portugal. This region is endemic for L. infantum, thus it is the ideal scenario to study if the ecological setting is consistent with natural exposure and infection.

What does prompt the analysis?

Authors’ response: The availability of samples from an ongoing project for the conservation  of the Iberian lynx (EU LIFE+ Project LIFE10NA-ES/570) an to evaluate if the available serological tools are adequate to address the Leishmania-specific seroreactivity.

We had recently set up the ELISA technique for cats and possess all the antigens in-house. Besides, we have performed similar analysis for dogs. Thus, it was a low-hanging fruit to respond to a question from the colleagues with our expertise.

Ultimately the generated data will contribute to increase the ecological information on these endangered felines.

Is it just exploratory or guided by any preliminary information?

Authors’ response: As stated above this was an opportunity study. There are previous studies that support infection in these animals. In this context, Iberian lynxe’s infection by L. infantum has been reported only once by Sobrino et al. 2008, who found 1 infected lynx among 4 studied animals. Since then, no further information has been produced in this regard. Thus, our study was based in this assumption and also in the available information on cats. It is exploratory in the sense that the ELISA cut-offs will probably be adjusted in the future after a clearer serological profile is established with access to more samples and ideally longitudinal observation of the study population.

Positive serology is obtained at very high titres only, meaning these may not be actual positive result. The serological methods seem to be not validated previously in this population of animals, which is understandable, but results should be taken with caution. Only 1 in 27 samples returned positive results for all tests used (1 molecular, 2 serological), is this really enough to state lynxes are susceptible to Leishmania infection? Also, serological positive results just coincided in one sample? This seems to be actually a limitation: small sample size and discordant results. 

Authors’ response: We thank the reviewer for the point made. We should state that seroreactivity is not the same as seropositivity. We do not have the means to establish disease-relevant cut-offs due to the limited number of animals and the absence of additional information. The cut-offs defined by us are more seroreactivity cut-offs. This was a major limitation of the work but also constituted a strength because we were not constrained by the need to adjust cut-offs to a clinical outcome. Even the IFAT cut-off was tentative and not based on the reported 1:80 cut-offs for feline leishmaniasis. Thus the proposed cut-offs are designed to answer the question: from which value do we consider that there are Leishmania-specific antibodies that are above the background level? To address this, we invested in three distinct Leishmania-specific antigens. SPLA, rK39, and CPX. We expect that Leishmania-specific responses would ultimately lead to an increase in the response to the three antigens. This was the case when using 1 cut-off. Still, we consider that there is also antigen-specific seroreactivity between 0.5 and 1 cut-offs. As stated this reactivity is probably more antigen-specific, similar to what is found in dogs from endemic areas without clinical signs (Lima C.S et al. 2022; doi.org/10.3390/ microorganisms10102018). The choice of SECA, a non-Leishmania specific antigens as a companion antigen contributed to strengthening the point on Leishmania-specific responses as was previously reported (Lima C.S et al. 2018; doi.org/10.1017/S0031182017000713). Thus considering these constraints, we do not think that the results are discordant. Looking at figure S4 and considering the high seronegative animals 6 out of 7 PCR-positive samples (that had serological tests combined) were associated with seroreactivity by ELISA. All IFAT-positive samples are associated with seroreactivity with some or all Leishmania-specific antigens. There is also a high level of agreement between seroreactive antigens. This is also supported by significant correlations in figures S5 and S6, especially between SPLA and rK39. Thus although the sample size is small, the objective of evaluating exposure/infection was achieved and we are now in the position to perform longitudinal evaluation of these animals for the ultimate validation of the proposed cut-offs.

Thus considering the specific question:

 “Only 1 in 27 samples returned positive results for all tests used (1 molecular, 2 serological), is this really enough to state lynxes are susceptible to Leishmania infection?”  

If this would be the case, we would say yes because it conjugated active infection with Leishmania-specific serology, this would be 3.7% of PCR-positive animals. Our most recent survey in dogs in presents 4.6% PCR-positive seropositive animals, this is not very different from the 1/27 proposed for the Lynx (Lima C.S et al. 2022; doi.org/10.3390/ microorganisms10102018). Still, the data is more complex than this. There is a significant overlap between PCR positivity and relevant seroreactivity suggesting that the overall conjunction of PCR and seroreactivity might be as high as 28%; Even if the cut-offs determined are upheld strictly, then three PCR-positive samples (3/8) were positive for at least one Leishmania-specific ELISA antigen and 3 of the IFAT-positive samples were PCR-positive.  

“Also, serological positive results just coincided in one sample?” 

Although the simple analysis of table 1 would suggest yes, in fact in figure S4 it is clear that several samples are seropositive to multiple antigens, and also all the IFAT-positive samples are also seroreactive to the Leishmania-specific ELISA antigens.

Sample. Is this a conveninece sampling taking advantage of the surveillance and monitoring program by ICNF? Or was sampling designed ad hoc for assessing prevalence of Leishmania infection in the lynx population? Please, clarify. 

Authors’ response: The samples were not collected for the purpose of this study alone. The analyzed samples were made available by ICNF for the purpose of this study and represent each of the lynxes captured during 2018 – 2020 as part of the ongoing surveillance and monitoring program carried out by ICNF under the framework of the EU LIFE+ Project LIFE10NA-ES/570. Therefore, the samples arose from convenience sampling. Nonetheless, a random minimum representative sample would have produced a smaller sample size considering the overall size of the population.

Please provide more details in how the sample was selected, the animal captured and handled during the process. And, where 2 mL collected from these animals (endangered species) just to assess whether they were exposed or infected by Leishmania? Was this accepted by and ethics or animal welfare committee? Please develop this aspect

Authors’ response: the sample of lynxes was taken from the animals as part of the surveillance and monitoring program implemented by the Institute for the Conservation of Nature and Forests (ICNF). The captures were performed within the framework of the EU LIFE+ Project LIFE10NA-ES/570). Animals were captured in the wild and handled under the supervision of veterinarians. Once a year, before the mating season, the ICNF veterinary experts proceed with the monitoring of the animals living in the reintroduction sites. For the purpose of biometric analysis, renewing identification collars, health evaluation, and disease surveillance, among others, animals must be trapped, sedated, and released after the necessary procedures were performed. The group studied in this research included the animals captured and released for the purpose of population health monitorization between 2018-2020. Again, we emphasize that the purpose of the sampling is related to the ongoing and necessary monitorization of the reintroduced and wild-borne lynxes, within the framework of the EU LIFE+ Project LIFE10NA-ES/570, under the welfare and ethical conditions determined by the panel responsible for this species conservation. Leftover sample material was used for the specific serological and molecular investigations presented in the manuscript. No blood was not collected on purpose for this study, results were useful for the clinical characterization of the lynxes. An ethical approval has been institutionally waived in this case, as the procedures were those of veterinary clinical assistance to the animals.

DNA used. DNA extraction started from 200 uL EDTA blood, please indicate elution volume and elution solution.

Authors’ response: The DNA extraction was performed from 200 µl of peripheral  blood preserved in EDTA and eluted in 100 µl of water for molecular biology use.

ELISA. Anti-cat IgG antibody is used, but not anti-lynx IgG. I understand the anti-lynx antibody may not be available, but I would take this as a reason for caution in the interepretation of the results. Is there any information available supporting the equivalence of anti-cat and anti-lynx IgG? If not, caution with the results.

Authors’ response: Reviewer 2 is again right in her/his remark. Indeed, that anti-lynx is not available so far. We do not have information on the equivalence of anti-cat and anti-lynx IgG. This is a polyclonal antibody and probably the immunoglobulins are sufficiently conserved to preserve recognition. In fact, the Leishmania-specific seroreactivity was compelling evidence that the antibodies detected were relevant and the assay was detecting Lynx antibodies. Still, a Lynx-specific secondary antibody if available would be the best option.

Calculation of seropositivity cut-offs was performed using the average seroreactivity of the 75% percentile plus 2x standard deviation" Please, explain the rationale for this.

Authors’ response: In the absence of a pre-defined positivity cut-off or the possibility to establish a cut-off based on the seroreactivity produced from lynxes known to have never experienced infection by L. infantum, the authors followed a population approach to define the positivity cut-off. For this purpose, the authors assumed that approximately 25% of the studied population could have been exposed to L. infantum (based on an overall seroprevalence prediction for CanL in the Mediterranean region by Franco et al. 2011;   10.1017/S003118201100148X) and therefore considered the Q75 as being a seronegative group. Then, to calculate the cut-off with 95% CI, the average seroreactivity of the Q75 (in this case, the average results equaled the mean results for each antigen) was added to 2xSTdev.  

IFAT. Negative control: wasn't it possible to use sera from a lynx from an area non-endemic for leishmaniasis?

Authors’ response: Reviewer 2 presents an interesting suggestion, but we did not have that kind of sample available. Therefore, it is not possible to obtain samples from Iberian lynxes from non-endemic regions in Portugal or Spain since the regions inhabited by the Lynxes are endemic for L. infantum.

Same comment as above regarding anti-cat IgG. These are limitations that point towards caution regarding the interpretation of the results.

Authors’ response: Reviewer 2 is again right in her/his remark. This point has already been addressed above. Anti-lynx conjugate is not available so far.

Accession numbers DNA sequences. Please, explain the rationale for using the following sequences and no other available in GenBank, AJ634344 (MHOM/PT/00/IMT260), AJ634343 (MHOM/ES/86/BCN16), AJ000294 247 (MHOM/CN/00/Wangjie1), AJ634361 (MHOM/SD/62/3S), AJ634370 248 (MHOM/SD/97/LEM3472)

Authors’ response: The DNA sequences were compared to those reported from L. infantum endemic regions within the Iberian Peninsula (such as Portugal, MHOM/PT/00/IMT260, and Catalunia MHOM/ES/86/BCN16) and to more distant countries such as China (MHOM/CN/00/Wangjie1) or   South America, since all these regions are endemic for L. infantum. Moreover these are sequences representative of different ITS-1 types as reported elsewhere (Kuhls e al. 2005, doi:10.1016/j.micinf.2005.04.009; ref. 44 of the manuscript).

Did the authors submit the DNA sequences they generated to the GenBank? Is there a plan for doing this? THis is good practice when reporting results related to DNA sequences.

Authors’ response: We plan to deposit the sequences associated to the ITS-1. The process of generating GenBank references has been started but is not yet finished. They will be deposited and associated to the publication.

Discussion. Please refer results to the specific study sample, this cannot be generalized to the Iberian lynx population unless the study design address this specificially.

Authors’ response: text in the Discussion section has now been adapted as well at the suggestion of Reviewer 1. We toned down the impact of the study to refer specifically to the 41 animals and not to generalize to the population. For example in line 383 of revised manuscript now reads: “we report the first survey of L. infantum infection involving 41 animals from a population of reintroduced and wild-borne Iberian lynxes living in the Portuguese Guadiana Valley Park”.

Discussion. "In the same report, restriction fragment length polymorphism (PCR-RFLP) analysis of the infecting Leishmania strain was unrelated to the ones circulating among canids from the same region, but similar to the pattern found on a L. infantum strain infecting a fox from a distinct geographical point." What is the purpose of this sentence? Is that the authors want to distinguish a sylvatic from a domestic cycle? Was that study designed to address that question?

Authors’ response: The reviewer makes a poignant point. In fact the study was not designed to address this question. The study was designed as, stated above, to evaluate the presence of Leishmania spp. in the study population of Iberian Lynx’s and also to address the capacity to detect Leishmania-specific antibodies by ELISA using a combination of Leishmania-specific and non-specific antigens. The mention of the strain found in a fox is just to highlight the existence of circulating strains in sylvatic and domestic cycle.  To make this connection more evident, we added to the section: “This highlights the existence of circulating Leishmania strains in sylvatic and domestic environments and the importance of typing of these strains to better understand the ecology of the parasite”

Discussion. "Due to DNA degradation and small volume, ITS-1 amplification was not possible in all kDNA PCR positive samples." So DNA degradation affected the performance of ITS-1 PCR but not that of kDNA PCR? And what do the authors mean by small volume? 2 mL blood were collected from the animals, and 200 uL used for DNA extraction, meaning this could have been repeated 10 times allowing ample volume for PCR testing. Besides, samples were stored at -20ºC, meaning good storage practice to avoid DNA degradation. Could it be possible to get more details on the timings and process of sample collection and preparation for analysis to understand where degradation could have occurred?

Authors’ response: The blood collection was not performed for the purpose of this study alone, and the current results have been produced in the context of a collaboration. The same samples have been used for other research purposes and the reminiscent of those samples were used in this screening. For the majority of the samples the authors had access to volumes of DNA that enabled no more than 3 PCR runs.

The outcome for amplification of Leishmania kinetoplast DNA is often superior, as the kinetoplast minicircle contains over 10,000 copies per cell. This fact makes this structure a good candidate for Leishmania molecular screening. On the other hand, ITS (internal transcribed spacer region) is located between the small subunit and the large subunit or ribosomal RNA genes, therefore presented in a much lesser abundance. Different detection thresholds have been reported for these Leishmania PCR targets depending on the protocol and analysed tissue, but overall ITS-1 targets present a higher detection threshold compared to kDNA primers (Leishmaniasis. Trends in epidemiology, diagnosis and treatment. Intech. Edited by David Claborn, 2014; Pag. 164; ISBN 978-953-51-1232-7)

Discussion. "... no data is available on the health background of these animals." How is this possible? Specimens from an endangered species are captured to assess Leishmania infection and no data is registered regarding their health status? The animals were not explored for signs compatible to leishmaniasis?

Authors’ response: For the purpose of this study, data on demographic parameters (ex. age and gender) and other health-related findings (ex. physical examination findings, hematological and biochemical findings) were not made available. We received full anonymized samples on our part. The compilation of that information will not be performed in the context of this publication. All the questions addressed by the reviewer in this section have merit but, as stated above, this was opportunity sampling and the main drive of the original study was not to assess Leishmania infection. Thus, the information obtained was not directed toward the visualization and registry of signs compatible with leishmaniasis. As stated above the overall aim of this work is to address if reintroduced and wild-borne lynxes are being infected by Leishmania spp. To evaluate this, we explored different molecular (PCR) and serologic approaches, the latter directed towards a better understanding of our capacity of for detecting anti-Leishmania antibodies in this animal species.   Proven this, the future work will involve longitudinal evaluation of these animals and monitoring their parasitological and serological status towards leishmania, plus health condition.

Discussion, line 433. Correct "co-mobilities"

Authors’ response:  corrected to read as “co-morbidities” (know found at line 120 of the Introduction)

" Recently, experimental evidence that Leishmania infection can occur by means of oral and intragastric transmission has been produced [52]. Based on this premise, since hares and rabbits (the major prey source of the Iberian lynx [31]) have been found infected with L. infantum [53–55], predator-prey interaction should be considered as a possible route of infection for lynxes." Is not more reasonable to consider that Leishmania infection would occur by the bites of infected sandflies? And in case this is true, will this be a transient or a persistent infection? Will the latter return more consistent results in terms of concordance between serological and molecular tests?

Authors’ response: Reviewer 2 is right in her/his remark. In fact, in areas of endemicity, infection by the bite of infected female phlebotomine sand flies is always the most likely mode of transmission. Nevertheless, other complementary modes of infection have been described. As mentioned in the manuscript, in a recent publication from Reimann et al. 2022, the authors concluded that golden hamsters who were infected with L. infantum via intragastric inoculation of infected macrophages (by gavage) had become infected. Leishmania was later isolated from the spleen of the infected hamsters by means of culture, besides DNA identification from intact skin, spleen and liver, and positive serology (IFAT titer 40). This way, infection and seroconversion was evidenced.  Besides the topic require further and more in-depth investigation, the authors pointed oral transmission as a possible infection route in mammalian leishmaniosis, similar to other Trypanosomatidae. Generally speaking, the question of whether Felidae natural infection with Leishmania turns into a transient or permeant infection remains a conundrum. Independently of the infection route, there is not enough scientific evidence or knowledge on the robustness of feline natural and cellular immune mechanisms that make these species less vulnerable to leishmaniasis and other VBD (Day, 2016; doi.org/10.1186/s13071-016-1798-5). Nevertheless, it is clear that rabbits and hares, who are the major food source of lynxes, are vulnerable to L. infantum and serve as reservoirs of this parasite. From the above, it is reasonable to consider the hypothesis of alternative and complementary routes of L. infantum infection in Iberian lynxes and other lagomorph mammalian predators. Reviewer 1 considered this section inadequate for the discussion because it would not help clarify the study, thus most of this information was adapted to be part of the introduction in the context of alternative means of infection.

 Conclusions "Our findings support evidence that this endangered species is susceptible to infection with L. infantum." This is a strong statement, maybe an important limitaton to indicate is the lack of concordance between tests, plus the inability of demonstrating the parasite in tissue.

Authors’ response: In this study, the evidence of infection was supported by the identification of Leishmania DNA in the blood of these animals, further supported by identification to the species level through complementary molecular techniques, besides the identification of circulating antibodies against Leishmania. Since L. infantum is the only proven endemic species of Leishmania in the Iberian Peninsula, the authors concluded that lynxes living in the Guadiana River Valley Park are susceptible to becoming infected by this parasite. It is not stated that these animals suffer from leishmaniasis, due to the fact that clinical evaluation of the studied lynxes was not performed for this purpose. In fact, it is concluded in this report that further biological relevance of these findings should be supported by a longitudinal follow-up of the studied animals, accounting for clinical evaluation besides serological and parasitological status of the animals towards Leishmania. As clarified above, we do not believe that such a significant lack of concordance exists between the tests. Still the discrepancies might be attributed to the gap between direct and indirect diagnostic test results can be justified, among other aspects, by: 1) the timeframe required to seroconvert; 2) the possibility of some of the infected animals never undergone seroconversion, as in dogs; 3) the possibility that seropositive animals have a false negative blood PCR result. Previous data support that bone marrow, lymph nodes and spleen are tissues harboring a higher parasite burden and therefore should be elected for PCR analysis. However, sampling of such tissues is invasive and not suitable to be performed in field conditions. Therefore, it is not adequate or in line with the scope of the ongoing conservation program for the Iberian lynx. The Discussion section was adjusted to better address the possible causes for some lack of agreement between serological and molecular tests (PCR).

Supplementary materials. Maps illustrating capture points would benefit of overlapping information on Leishmania presence, if available. Otherwise it should be explained whay those capture areas were selected if information on Leishmania presence is not available. 

Authors’ response: We thank once again to the reviewer on the attention to detail associated to the review. The Guadiana River Valley Park was selected because it is the only site for Iberian lynxes reintroducion. No other place in Portugal has a stable population of these animals. Thus it was a default selection, as there are no other equivalent sites in Portugal. The park is situated in the south-eastern part of the Alentejo region. There is no detailed serological information on Leishmaniasis in this particular scenario of the Guadiana River Valley. Even if it existed, the foraging areas associated to the animals probably include other regions.

Round 2

Reviewer 2 Report

The authors have addressed comments made by the reviewer and modified the text accordingly when necessary. In my opinion, the manuscript has improved in terms of clarity. I'm staified with the review work conducted by the authors.